# Deer Slayers: Examining the Scope of and Arguments for and against Legal Deer Theriocide in the US

**Michael J. Lynch** [1],[*] and **Leonard J. Genco** [2]

1 Department of criminology, University of South Florida, Tampa, FL 33620-8100, USA
2 Criminology and Criminal Justice Program, University of Tampa, Tampa, FL 33606, USA
* Correspondence: mjlynch@usf.edu

**Abstract:** Deer hunting has a long history in the US. It is supported by hunting cultures, described as necessary for protecting forest/plant biodiversity and ecosystems, but opposed by animal welfare and rights advocates as cruel. Using multiple literature sources, we examine the trade-off between protecting deer and ecosystems from harm in the context of contemporary America. We examine various approaches for exploring harms affecting nonhuman animal populations found in the green criminological, environmental sociology, wildlife conservation and management, and ecological literature. We argue that making sense of these opposing positions requires examining the extent of deer hunting to quantify those harms in some way. Here, we examine reported deer kills for US states for the period 1999–2020. These data indicate that nearly 7 million deer are taken annually in the US. We also examined some hypothesized correlates of deer harvesting across states. While these data tell us something about the number of deer killed, these data alone are insufficient. We argue no clear conclusion about the acceptability of deer hunting can be reached given the difficulty rectifying opposing moral/philosophical positions on deer hunting, opposing deer management objectives, and scientific evidence on the ecological impacts of deer populations in the US under contemporary conditions that include shrinking forest ecosystems and impaired ecosystem stability.

**Keywords:** deer hunting; deer management; US deer killed legally; theriocide; issues in the control of deer populations; green criminology; effects of deer on ecosystems; conservation criminology





## 1. Introduction

Deer hunting has a long history in the US. The title of our article includes the term "deer slayer," which was the name of an 1841 novel by early American writer, James Fenimore Cooper, recognizes that long tradition. This term is not meant in a derogatory sense but is included to recognize the long hunting heritage in the US. Deer hunting has been supported by hunting cultures and deer hunting licensing regulations [1], and as a mechanism that contributes to biodiversity and the conservation of nature [2,3]. Numerous groups, however, oppose deer hunting, viewing it as immoral and a violation of animals' rights [4,5]. Debates concerning the morality or necessity of hunting—whether for recreational, conservation or subsistence purposes—can involve appeals to philosophical arguments concerning the morality of hunting, or those concerning the right to hunt [6,7]. Whether these debates are philosophical/moral or scientific, there is a need to quantify the extent and type of harm occurring, though some might argue that moral issues are absolute and apply even when only one animal is harmed, even minimally. Nevertheless, discussions of the scope of harm caused by hunting requires some discussion of data that can be employed to depict those harms as well as their potential positive outcomes (e.g., subsistence food source; limiting ecological damage).

The primary focus of this manuscript is examining the extent of deer killing by deer hunters in the US. Deer killing can be defined in opposing ways from different perspectives. In moral/philosophical views, hunting is viewed negatively, while in other perspectives,

deer killing may be seen positively (e.g., for a hunter), or interpreted in a more neutral manner (e.g., in deer population management views). It is not our goal to take up any of these viewpoints, but to discuss how counting deer killed by hunters has relevance to those approaches. To do so, we examined data on legally reported deer killings by hunters to US state agencies (except Hawaii) for a twenty-two year time period (1999–2020). These data have not been previously examined in the extant literature. While we employ the term "deer killing" frequently below, we do not intend this term to infer a negative image, but find there are few other terms that can be substituted (e.g., deer taking or harvesting). Each state has a system for the purchase of deer hunting licenses, and for counting the number of deer taken legally during hunting season. In the US, approximately 7 million deer are reported as legally taken each year (see below). Although the number of illegally killed deer in the US is unknown, it is likely a fraction of the number of deer killed legally [8].

The background context for this analysis draws from literature in several fields. In recent years, for example, green criminology—the branch of criminology that addresses crimes and harms affecting nature, and may involve the study of both illegal and illegal harms and questions of justice that are philosophical and/or involve the enforcement of environmental law—and researchers in other fields (e.g., [9,10]) have drawn attention to the morality of animal killings using philosophical arguments. Less attention has been directed toward the empirical extent of this issue such as counting animal killings, including killings classified as legal or permitted by law, and examining how that data intersects with initiatives to protect ecosystems from harm. In examining these literatures, one of the most basic questions one can ask is: how many deer are harmed? Currently, there is no answer to that question in the extant literature.

We begin with background concerning the study of animal killings and briefly review arguments that: (1) evaluate deer hunting as necessary for the maintenance of deer populations and/or ecological spaces; (2) view hunting as a right; and (3) those that argue that deer killing (i.e., theriocides) should be banned due to the harms caused to animal populations or individual animals. Next, we examine reported deer kill data, and the scope of legal deer killing in the US. Those data indicate a good deal of deer killing. Our discussion of those data examines deer killing in relation to the various deer hunting positions described in the background section.

## 2. Background: The Study and Scope of Animal Killings

Humans kill numerous animals in large numbers through activities such as food production, farm animal and pet abuse, wild animal population management, fishing and hunting, in the breeding of laboratory animals, and carrying out laboratory animal research. In 2014, Beirne [11] introduced the term theriocide to identify "those diverse human actions that cause the deaths of animals" [11] (p. 55). He added that "As with the killing of one human by another, a theriocide may be socially acceptable or unacceptable, legal or illegal" [11] (p. 55). In short, animal killings (theriocides) are not unidimensional, and there are many types of legal and illegal theriocides. This observation leaves open the possibility that different forms of theriocide ought to be addressed in unique ways. Consistent with this observation, several studies by green [12–15] and conservation criminologists [16,17] have drawn attention to various legal and illegal forms of animal killings.

The related criminological literature examines the definition of crimes against animals [18,19]; the ways in which animals are killed, legally and illegally [20]; and the social control of crimes against animals [21–23]. In this way, criminology has created a space in which crimes/harms affecting animals can be addressed. Some argue that this has been less true in the sociological literature. For example, Carter and Charles [24] note that sociology's speciesist orientations has excluded animal issues, and instead the inclusion of animals within sociology requires adopting the view that animals are involuntarily embedded in human social relations (see also [25]). Regardless of the discipline or theoretical orientation, few studies explore the volume of animal harm/killings that occur in society as an important aspect of examining harms against animals. Understanding the scope of animal

killings provides important information for interpretation of human–animal interactions, for conceptualizing how humans perceive and value animals, the level of harm to which animals might be subjected, and, as some argue, how perceptions of animals affect the socially constructed nature of the relationships between animals and humans [20].

*2.1. Counting Harms against Animals*

Counting the number of animals humans harm or kill is no easy task. Harms against animals are widespread in society and include the violent victimization of farm, laboratory, and companion animals [18,22]. Some of these topics are widely explored outside of criminology (e.g., on laboratory animal harms, see [26]; for a review and classification of the related literature, see [27]). One reason these harms may be glossed over involves a lack of relevant data concerning counts of animal harms. It is difficult to produce those counts for several reasons. First, humans harm animals in numerous ways, and those harms include a wide array of animals. Data on the killings of all kinds of animals may not be tracked or published. Second, counting animals harmed/killed in specific ways is sometimes difficult because of a failure to perceive those harms/killings as outcomes that constitute a behavior that includes a harm against animals. This might include, for example, counting the number of animals used in food production, or in scientific experiments. Third, there may be no entity assigned the task of counting those behaviors. Put another way, social definitions of harms may not be sensitive to harms against certain animal species (e.g., the hunting of predator or nuisance animals), while it may be more sensitive to other species (e.g., ecological keystone species), affecting official counts of harms against animals. Finally, while there are some databases that count animal killings, those counts are not centralized, and likely are not reliable. Consequently, valid counts of the number and types of animals harmed and killed are difficult to locate.

To explore one aspect of animal killing that also involves a debate concerning the morality and social utility of those killings, this article examines the scope of legal deer killing in the United States. Deer hunting is one of the many forms of harms humans can legally (and illegally) commit against animals. There is a broad body of literature promoting the killing of deer as a necessary tool for managing deer populations [28]. Yet, there is little discussion that frames deer hunting as a harm in the academic research literature generally [29], or in the criminological literature in particular [30], and little effort has been made to ascertain the extent of legal deer taking in the US.

*2.2. Prior Research*

As noted, the study of harms against animals is a growing area of research in criminology, promoted by green criminologists, and linked to the groundbreaking work of Piers Beirne [31]. Building on literature in other disciplines, Beirne [11] introduced the term theriocide to the criminological literature to identify both the legal and illegal killings of animals. Beirne's argument suggests that the killing of animals deserves a specific term to recognize the importance of those killings, just as we recognize the killing of humans by using the term "homicide." Since its introduction, this term has become widely employed. The majority of studies on theriocide are theoretical or involve qualitative information and case study approaches [12,32–34]. Though studies of animal killings do not always refer to the term "theriocide" in their title, several green and conservation criminological studies have examined animal killings empirically (on wildlife killings in Sweden, see [35]; on officially sanctioned wildlife killings in the US, see [13,36]).

Environmental sociology suggests that society's economic orientation contributes to the killing of animals. Specifically, those studies examine the declining concentration of wildlife species, noting that the killing of species/animals has increased along with the expansion of capitalism and processes within capitalism such as the treadmill of production. For example, Shandra et al. [37] examined species losses for mammals and birds across 65 nations and found support for arguments from dependency theory suggesting that higher debt levels in developing nations lead to increased resource extraction to pay those

debts, and to increased species loss (see also [38]). In an earlier study, Hoffman [39] found that species endangerment was related to world-system position, measures of the treadmill of production, and ecological modernization for a sample of 120 nations. Clausen and York [40] suggested that these structural processes also explain the long-term trend in fish stock loss. Similar economic arguments have been applied to wildlife hunting, enclosure movements designed to commodify wildlife and hunting access [41], and the promotion of a hunting industry [42].

The criminological literature has employed observations from environmental sociology to explore species loss concerns theoretically [43]. Little of the criminological literature addresses the subject of this paper—deer killing. Extant research on deer taking examines the historical origins of deer poaching laws [30], while a handful of studies examine causes of illegal deer hunting, and typologies of illegal deer hunters [44–46]. To date, studies have overlooked counts of deer killings and trends in those counts, especially as indicators of the extent of theriocide.

### 2.3. Tracking Deer Killings

At various points in history, deer hunting had large impacts on deer herds in various states (see below), and earlier in history deer populations were controlled by large predators before introduction of predator bounties in the US. The number of deer hunters can impact deer population size. On average over the past twenty years, there were approximately 11 million gun and bow deer hunters in the US, who killed an average of 6.7 million deer/year. Early in the 20th century, recognizing that deer hunting may have adverse impacts on deer herds, some US states began tracking deer hunting. For example, California (1911), Arkansas (1938), Maine (1919), Pennsylvania (1915), and Vermont (1897) have fairly long official deer kill counts. States collect these data to track and manage the deer harvest to maintain a "healthy" deer population (see discussion below). Deer hunting and conservation research suggests it was deer hunters' interests in maintaining healthy wild deer herds that effectively saved deer from possible extinction following extensive subsistence and big game hunting [47]. Here, use of the term "healthy" has interesting implications.

In the deer hunting literature, the term "healthy" does not refer to species or ecological health from the perspective of deer. Rather, when states collect data on deer hunting, they are interested in maintaining a healthy deer population that provides sufficient opportunities for hunters, while not endangering the state's deer population to the extent that it becomes decimated [8]. Maintaining healthy deer populations can also contribute to the economic survival of large landowners who rent access to privately owned farms and forest areas to hunt deer [48] and has implications for the hunting sectors of the economy [49].

Beginning in the 1970s, conservation managers attempted to respond to a wider variety of stakeholder concerns to redefine the term "healthy" [50], that included recognition of different deer hunting cultures/interests [8,51,52]. Since that time, research has endeavored to devise methods for evaluating the effect of public participation in the management of deer herds. In the academic literature, three alternative perspectives evolved to theorize about, measure, and increase public participation in environmental decision making that affected deer populations [50]. An important observation here is that different stakeholders have different concerns. Consequently, when a conservation effort is launched or is strongly influenced by a particular stakeholder group, the outcome is likely to favor that group. Take, for example, the formation of a deer management cooperative (DMC). The purpose of a DMC is to coordinate private landowners' and hunters' interests in improved deer herds that attract hunters. Through DMCs, parties voluntarily cooperate to enhance wildlife habitat and the "hunting experience" [53]. Deer, not being able to represent themselves, are not considered to be stakeholders in these kinds of conservation management approaches. Rather, the deer are treated as a public resource, and valued as a hunting commodity and for the hunting "experience" they provide. In other cases, deer are perceived as a nuisance, and strategies are sought to constrain the human–wildlife conflicts deer present, particularly in suburban neighborhoods [54], which includes proposals for commercial deer

harvesting [55]. In contrast, in some countries such as Japan, urban residents have "more positive" attitudes toward deer for aesthetic reasons and seek to manage deer populations outside of hunting [56]. These observations imply that the salience of deer hunting cultures in different countries or within regions within countries may affect attitudes toward using hunting as a deer management tool [57]. In some cases, attitudes toward deer management are also affected by the extent to which deer impact rare plants [58]. Differences in the composition of regional stakeholders, conservation goals, human–wildlife conflicts, and public attitudes have led US states to adopt different policies concerning the management of deer populations [59]. Moreover, in the current context of deer overabundance in many US locations, the question has been raised whether deer hunting is sufficient to control expanding deer herds and the environmental harms they cause [8,60], and has also been raised as a response to controlling deer tick infestations [61–64].

In sum, the primary impetus behind state level deer hunting databases has been monitoring the health of deer herds to promote "fairly" regulated hunting opportunities for the public that will not lead to the annihilation of deer herds within a state [8]. As an example, consider the trend in deer killings and deer hunting regulation in Ohio observable in the data collected for this study. Ohio has maintained deer hunting data since 1901. During the early 20th century, some argued that by 1904 "Ohio was without whitetail" [65] (p. 12). In response, Ohio banned deer hunting on several occasions to repopulate deer herds. These bans occurred from 1901 through 1942 (42 years), and again in 1946, 1949, 1951, 1953, 1954, and 1961. Thus, for the sixty-one year period (1901–1961), deer hunting only occurred during thirteen years, and was banned in 48 years. Thereafter (1962–2020), deer were regularly hunted in Ohio. After 1962, Ohio reported 6,503,031 deer killed, an average of 110,221 deer per year, with a low of 406 in 1965, and a high of 261,314 in 2009. Four different periods of deer hunting can be discerned based on the average number of deer killed in Ohio: (1) 1962–1973, about 2568 deer were killed per year; (2) 1974–1987, mean 43,446 deer killed/year, a about 17-fold increase over the first period; (3) 1988–2001, mean 135,000 deer killed/year, a threefold increase compared to period 2; and (4) 2002–2020, mean of 208,840 deer killed/year, when deer killing became more consistent from year to year. By period 4, deer killing had increased by a factor of 81 compared to period 1.

Although it is not our intention to analyze deer killing trends for each state individually, the Ohio example illustrates how the quantity and trends for deer killing within a state can vary over time. Also, these data suggest that state management of a deer population can facilitate the growth of a deer herd large enough to support sport hunting. That latter observation implies that sport hunting-related conservation can preserve a species existence locally. That possibility makes it difficult to frame deer hunting in an entirely negative manner. Conversely, it should be noted that deer management has increased deer abundances in many states, producing an *unintended negative effect* that has caused shrinking forest ecosystems or damage to commercial forest enterprise [41,66–69]. Numerous studies detail the adverse impact of deer on forest ecosystems [70], and those observations make it difficult to dismiss the need for some mechanism for managing the size of US deer herds. It is beyond the scope of the current discussion to suggest whether deer management should or should not include deer hunting or other options such as commercial harvesting, reintroduction of predators, and contraceptives to control deer populations and preserve ecosystem stability [28,59,71–74]. Also, given that the management of deer herds increases their size and hence the opportunity for hunting, deer management has increased the number of deer killed. In some philosophical views, one could potentially argue against deer management, then, as a cause of suffering for deer—a question we have not seen addressed in the related literature. It should be noted, however, that studies indicate that significant attention must be paid to wildlife harvesting policies since those policies can lead to the collapse of harvested populations [75].

## 3. Materials and Methods

Data for this study represent known/reported deer killings in each state from 1999 through 2020 (22 years). Data for deer killings in Hawaii were excluded, since the deer population hunted there are dissimilar to those hunted in the rest of the US, and herd size is rather small. Data for the entire period was not available for every state, but all states reported 19 or more years of data as follows: 19 states reported 22 years (38.8%); 18 states reported 21 years (36.7%); seven states reported 20 years (14.3%); and five states reported 19 years (10.2%) of data.

States collect deer hunting data using different methods. Each method has limitations with respect to complete reporting of deer kills. These data are, as noted below, either self-reported, or collected by state field agents. Given these methods, it should be excepted that deer kills are underreported.

The most popular state data collection methods include mandatory field reporting (N = 25), and post-hunting deer hunter surveys (N = 14) that consist of online, mail-in, and telephone surveys. The remaining states (N = 10) collect data using field check and road check stations. The quality of data collection varies across states. Prior studies indicate that certain methods provide better deer kill count estimates [62,63,76,77]. Some states (e.g., Florida since 2015) were known to have reporting difficulties due to budget cuts that resulted in declining deer kill reports. Given that our interest is estimating the approximate extent of deer killed annually and not in providing an explanation for changes in reported deer kills in a given state over time, these reliability and validity measurement issues imply that our kill estimate is imprecise but should be valid enough for the purposes of our assessment, which is estimating the approximate number of deer killed in the US annually and over the study period. It should be noted, however, that we would expect underreporting of deer kills, even though some states adjust their reports to reflect under-reporting using various methods that are not uniform across states.

Raw numbers tell part of the story about deer harvesting. Kill rates standardize the number of deer taken and make them more comparable across states. We created kill rates per 100 population of people in each state since there are no reliable or valid estimates of deer herd size in each state for every data year in our study. Alternatively, these estimates could now be made using recently released (October 2022) hunting license data from the US Fish and Wildlife Service that begins in 1959. These data could be employed to create a deer kill/hunting licenses sold ratio. These recent data could be useful if research was exploring the possible effects of hunting culture prevalence on deer taking. The limitation of these data is that the license sold information is not species-specific, and deer are not the only species hunted, especially in certain locations. Since our study covered the period 1999–2020, we used the mean number of deer killed and the mean human population for each state for the 1999–2020 period to calculate the deer killed rate: ((mean deer kills in a state, 1999–2020)/mean human population in a state, 1999–2020)) × 100.

In addition to using these data to examine the distribution of deer taking within and across states, there data were also employed to create deer hotspot/cold spot maps, and to determine whether a few potentially important predictors of deer taking could be identified. Further details on these procedures follow.

### 3.1. Deer Kill Hot/Cold Spot Maps Materials and Methods

Another way to examine the deer taking data is through the use of hot and cold spot maps. Maps examining the distribution of the data based on standard deviations were generated with GeoDa (https://geodacenter.github.io/, accessed on 12 January 2023). Standard deviations from the mean are appropriate since other techniques used for dis-aggregated data such as the Getis–Ord are not recommended for units of analysis with fewer than eight close neighbors (see ArcGis handbook). Geographically, the majority of US states do not have eight close neighbors, and are bordered by oceans, the Gulf of Mexico, the great lakes, Canada, and Mexico. Following widely agreed upon statistical standards, standard deviations of 2 or more are considered statistically significant. On the

maps, however, we also indicate states that are one standard deviation above the mean. Map standard deviations for hot spots are shaded red and are darker for higher standard deviations. Standard deviations above 2 are also shown.

### 3.2. Deer Kill Distribution Predictions Methods

It is also useful to test some basic hypotheses about the distribution of deer taking to determine if that distribution can be predicted empirically. Here, we limit our analysis to a few preliminary potential predictors of deer harvesting rates across states and time. Given some of the limitation in the counting of deer kills, these results contain unknown measurement error (see earlier description of the data). Moreover, one would hypothesize that the most important correlate of deer taken would be deer hunting licenses. That data is currently unavailable, except as noted above, and would certainly be required if a multivariate model of factors affecting deer hunting across US states was estimated.

Historically, deer hunting played an important role in providing protein for subsistence. In contemporary America, there are relatively few deer hunters (about 4.25% of the US population), but widespread support for subsistence deer hunting with some studies indicating that 85% of the US population supports subsistence deer hunting [47]. Research indicates that in some US states, participation in hunting is a function of subsistence needs and income [78]. Moreover, it should be acknowledged that subsistence hunting is of particular importance to indigenous peoples [79]. The small percentage of indigenous peoples in the US, however, is not likely to have much effect on deer kill rates across states. Based on the above, we hypothesize a relationship between household income and deer hunting across states.

In addition to income, one could also hypothesize a relationship between gun ownership rates and deer hunting. Firearm ownership rates and hunting license rates, however, may not be correlated. Firearm ownership data were drawn from Rand (https://www.rand.org/pubs/tools/TL354.html, accessed on 20 June 2022).

For many decades (since the late 1920s), the sociological and criminological literature has discussed whether a subculture of violence might impact the distribution of violent and gun-related harm outcomes. In that view, some places are more prone toward violence and potentially related to circumstances such as gun ownership. Thus, in addition to gun ownership, it could be hypothesized that places with higher levels of violence may have higher deer kill rates. While there are many potential level of violence indicators, here we selected the gun homicide rate. Firearm mortality data were collected from the Center for Disease Control and Prevention website (https://www.cdc.gov/nchs/pressroom/sosmap/firearm_mortality/firearm.htm, accessed on 21 June 2022).

Given the preliminary nature of this research and the lack of other important data (i.e., hunting license data), we tested the above hypotheses in the simplest way possible using correlations between predictors and deer kill rates. The data for these assessments represent means across US states. Means were employed because data for the predictors were not available for each of the years for which we collected deer harvest counts by state. These results, therefore, are suggestive, and should not be interpreted as a definitive test of the above hypotheses.

## 4. Results

The data were examined in different ways to illustrate their relevance for estimating and assessing the number of deer killed. We present these data below in two primary sections on *raw deer kill counts* and *rates of deer killed*. In addition, sections addressing the preliminary results for the deer taking distribution and the hot/cold spots maps are found in this section.

### 4.1. Deer Killed Counts

Table 1 shows raw count outcomes. In Table 1, column "N Killed," is the number of deer taken in each state for the entire time period. The "Mean" column is the average

number of deer killed each year in a state. Column five and column six show the high and low number of deer taken in each state during the study period (1999–2020).

**Table 1.** Total number of deer killed by state, 1999–2020 (except as noted, where Nis < 22).

| State | N Years [1] | N Killed | Mean | High Kill N [2] | Low Kill N [3] |
|---|---|---|---|---|---|
| Alabama | 22 | 7,338,314 | 333,560 | 499,000 (2004) | 195,128 (2020) |
| Alaska | 21 | 322,307 | 15,348 | 20,529 (2005) | 9078 (2000) |
| Arizona | 20 | 252,251 | 12,613 | 20,313 (2017) | 9268 (2003) |
| Arkansas | 21 | 3,763,708 | 179,224 | 213,487 (2012) | 108,456 (2003) |
| California | 21 | 379,821 | 18,087 | 27,608 (2016) | 10,892 (2011) |
| Colorado | 20 | 730,918 | 36,546 | 45,000 (2007) | 27,492 (2014) |
| Connecticut | 20 | 237,302 | 11,865 | 13,541 (2004) | 9113 (2015) |
| Delaware | 21 | 291,005 | 13,857 | 17,265 (2020) | 10,300 (2002) |
| Florida | 21 | 2,372,903 | 112,995 | 180,000 (2008) | 68,827 (2019) |
| Georgia | 20 | 7,538,830 | 376,942 | 464,003 (2010) | 262,042 (2019) |
| Idaho | 21 | 989,624 | 47,125 | 68,764 (2015) | 22,337 (2001) |
| Illinois | 22 | 3,687,482 | 167,613 | 201,209 (2005) | 144,303 (2016) |
| Indiana | 21 | 2,501,659 | 111,127 | 136,248 (2012) | 99,618 (1999) |
| Iowa | 21 | 2,774,895 | 132,138 | 211,451 (2005) | 99,400 (2013) |
| Kansas | 22 | 1,829,855 | 83,175 | 111,159 (2000) | 71,283 (2003) |
| Kentucky | 21 | 2,613,612 | 124,458 | 155,730 (2015) | 95,229 (1999) |
| Louisiana | 19 | 3,322,628 | 174,875 | 250,000 (2000) | 120,800 (2018) |
| Maine | 21 | 560,972 | 26,713 | 38,153 (2002) | 18,045 (2009) |
| Maryland | 22 | 1,973,553 | 89,707 | 100,663 (2009) | 77,382 (2018) |
| Massachusetts | 21 | 243,117 | 11,577 | 14,551 (2018) | 9634 (1999) |
| Michigan | 20 | 8,670,442 | 433,533 | 544,895 (1999) | 334,612 (2015) |
| Minnesota | 21 | 4,385,131 | 308,815 | 290,525 (2003) | 139,442 (2014) |
| Mississippi | 21 | 5,583,753 | 265,893 | 322,287 (2010) | 194,975 (2017) |
| Missouri | 20 | 5,979,268 | 298,963 | 324,425 (2006) | 250,105 (2000) |
| Montana | 22 | 2,306,590 | 104,845 | 140,000 (2003) | 75,680 (2014) |
| Nebraska | 22 | 1,369,689 | 62,259 | 88,034 (2010) | 52,174 (1999) |
| Nevada | 22 | 117,337 | 6243 | 12,499 (2000) | 5831 (2011) |
| New Hampshire | 22 | 248,291 | 11,286 | 13,559 (2007) | 9143 (2001) |
| New Jersey | 21 | 1,177,750 | 53,534 | 77,444 (2000) | 41,439 (2015) |
| New Mexico | 20 | 214,303 | 10,715 | 15,500 (1999) | 7502 (2011) |
| New York | 22 | 5,146,702 | 233,941 | 308,216 (2002) | 180,214 (2005) |
| North Carolina | 22 | 4,416,958 | 155,316 | 188,130 (2013) | 118,174 (2002) |
| North Dakota | 19 | 1,236,689 | 65,089 | 109,676 (2006) | 32,702 (2014) |
| Oklahoma | 22 | 2,267,187 | 103,054 | 126,000 (2020) | 82,724 (1999) |
| Ohio | 22 | 4,410,012 | 200,455 | 261,314 (2009) | 126,770 (1999) |
| Oregon | 22 | 1,009,974 | 45,908 | 63,507 (1999) | 30,464 (2020) |
| Pennsylvania | 22 | 8,312,698 | 377,850 | 517,529 (2002) | 308,920 (2009) |
| Rhode Island | 19 | 44,182 | 2326 | 2937 (2008) | 1883 (2015) |
| South Carolina | 21 | 4,747,251 | 226,060 | 319,902 (2002) | 172,315 (2016) |
| South Dakota | 21 | 1,417,409 | 67,496 | 94,726 (2010) | 48,500 (1999) |
| Tennessee | 22 | 3,642,940 | 165,588 | 182,064 (2006) | 135,150 (2019) |
| Texas | 19 | 13,724,647 | 722,350 | 971,677 (2016) | 605,974 (2000) |
| Utah | 19 | 565,108 | 29,742 | 37,804 (2016) | 23,124 (2011) |
| Vermont | 22 | 355,057 | 16,139 | 20,498 (2000) | 8546 (2005) |
| Virginia | 22 | 4,757,130 | 216,233 | 259,147 (2009) | 180,121 (2016) |
| Washington | 22 | 733,732 | 33,352 | 44,544 (2004) | 27,187 (2019) |
| West Virginia | 21 | 3,096,166 | 147,437 | 255,356 (2002) | 99,437 (2019) |
| Wisconsin | 22 | 8,814,261 | 400,648 | 615,293 (1999) | 291,023 (2019) |
| Wyoming | 21 | 990,703 | 47,176 | 55,061 (2007) | 39,260 (2013) |
| Total | | 142,343,597 | | | |
| Mean | 21.04 | 6,765,380 | 138,069 | | |

[1] "N Years" is the number of years (maximum = 22) for which deer taken totals were reported for each state. [2] The "High Kill N" indicates the largest number of deer killed in each state, and the year in which that occurred. [3] The "Low Kill N" indicates the lowest number of deer killed in each state, and the year in which that occurred.

The total number of deer harvested was 142,343,597, with an unadjusted mean of 6.765 million/year (the adjusted mean for missing data is 6.85 million). The most deer were killed in Texas (N = 13,724,647) and accounted for 9.64% of all deer taken. Following Texas, three states reported total deer kills above 8 million: Wisconsin (8,814,261), Michigan (8,670,442), and Pennsylvania (8,312,698). These four states reported more than 39.5 million deer harvested, representing nearly 27.8% of all deer killed in the US. Two states, Alabama (N = 7,338,314) and Georgia (N = 7,538,830) round out the top cluster of high kill states.

The total deer killed in these six states was more than 46.59 million, representing 33% of all deer kills. The human population in those states comprises approximately 22% of the US population.

The next highest group of deer kill states included Minnesota (4,385,131), Mississippi (5,583,753), Missouri (5,979,268), New York (5,146,702), North Carolina (4,416,958), Ohio (4,410,012), South Carolina (4,747,251), and Virginia (4,757,130). In these eight states, about 39.42 million (27.7%) additional deer were killed. Taken together, in the top fourteen states, 86 million deer were harvested, representing 60% of all deer killed during the study period.

At the bottom end of the distribution, eleven states produced fewer than half-a-million deer kills during the study period. These states included Alaska (322,307), Arizona (252,251), California (379,821), Connecticut (237,302), Delaware (291,005), Massachusetts (243,117), Nevada (117,337), New Hampshire (248,291), New Mexico (214,303), Rhode Island (44,182), and Vermont (355,057). These states make up 22% of states in the data set, but only 1.7% of all deer killed (N = 2,443,073).

The state deer killed annual mean ranged from a low of 2326 in Rhode Island, to a high of 722,350 in Texas. Consistent with that result, the lowest annual kill total was in 2015 in Rhode Island (N = 1883), while the highest was in Texas in 2016 (N = 971,677).

### 4.2. Deer Kill Rates

Data on the kill rate can be found in Table 2 in column four. For ease of comparison, Table 2 also contains the number of deer killed by state in column 2. In that table, column 3 shows the number killed rank, column five displays the kill rate rank, and column 6 the average or mean of these two kill measures as the mean kill rank. In a later map, we show deer kill rates using deer population counts for 2017, the only year there are complete deer population estimates by state.

**Table 2.** Number killed (column 2), kill rank (column 3), kill rate (column 4), kill rate rank (column 5), mean rank for N killed and kill rate (column 6) by state, 1990–2020.

| State | N Killed | N Kill Rank | Kill Rate | K Rate Rank | Mean Rank [1] |
|---|---|---|---|---|---|
| Alabama | 7,338,314 | 6 | 7.07 | 8 | 7 |
| Alaska | 322,307 | 41 | 2.17 | 27 | 34 |
| Arizona | 252,251 | 42 | 0.199 | 47 | 44.5 |
| Arkansas | 3,763,708 | 15 | 6.16 | 9 | 12 |
| California | 379,821 | 39 | 0.005 | 49 | 44 |
| Colorado | 730,918 | 36 | 0.717 | 40 | 38 |
| Connecticut | 237,302 | 46 | 0.336 | 45 | 45.5 |
| Delaware | 291,005 | 43 | 1.55 | 34 | 38.5 |
| Florida | 2,372,903 | 23 | 0.605 | 42 | 32.5 |
| Georgia | 7,538,830 | 5 | 3.77 | 16 | 10.5 |
| Idaho | 989,624 | 33 | 2.88 | 20 | 26.5 |
| Illinois | 3,687,482 | 16 | 1.30 | 35 | 25.5 |
| Indiana | 2,501,659 | 22 | 1.85 | 29 | 25.3 |
| Iowa | 2,774,895 | 20 | 4.371 | 12 | 16 |
| Kansas | 1,829,855 | 27 | 3.07 | 18 | 22.5 |
| Kentucky | 2,613,612 | 21 | 2.868 | 21 | 21 |
| Louisiana | 3,322,628 | 18 | 3.86 | 15 | 16.5 |
| Maine | 560,972 | 38 | 2.03 | 28 | 33 |
| Maryland | 1,973,553 | 26 | 1.72 | 31 | 28.5 |
| Massachusetts | 243,117 | 45 | 0.175 | 48 | 46.5 |
| Michigan | 8,670,442 | 3 | 4.365 | 13 | 8.5 |
| Minnesota | 4,385,131 | 14 | 3.95 | 14 | 14 |
| Mississippi | 5,583,753 | 8 | 8.98 | 3 | 5.5 |
| Missouri | 5,979,268 | 7 | 4.74 | 11 | 9 |
| Montana | 2,306,590 | 24 | 10.70 | 1 | 12.5 |

**Table 2.** *Cont.*

| State | N Killed | N Kill Rank | Kill Rate | K Rate Rank | Mean Rank [1] |
|---|---|---|---|---|---|
| Nebraska | 1,369,689 | 29 | 3.44 | 17 | 23 |
| Nevada | 117,337 | 48 | 0.304 | 46 | 47 |
| New Hampshire | 248,291 | 44 | 0.852 | 39 | 41.4 |
| New Jersey | 1,177,750 | 31 | 0.642 | 41 | 36 |
| New Mexico | 214,303 | 47 | 0.527 | 43 | 45 |
| New York | 5,146,702 | 9 | 1.195 | 36 | 22.5 |
| North Carolina | 4,416,958 | 12 | 1.64 | 33 | 17.5 |
| North Dakota | 1,236,689 | 30 | 9.46 | 2 | 16 |
| Oklahoma | 2,267,187 | 25 | 2.77 | 23 | 24 |
| Ohio | 4,410,012 | 13 | 1.74 | 31 | 22 |
| Oregon | 1,009,974 | 32 | 1.190 | 37 | 34.5 |
| Pennsylvania | 8,312,698 | 4 | 2.99 | 19 | 11.5 |
| Rhode Island | 44,182 | 49 | 0.220 | 47 | 48 |
| South Carolina | 4,747,251 | 11 | 5.03 | 10 | 10.5 |
| South Dakota | 1,417,409 | 28 | 8.23 | 5 | 16.5 |
| Tennessee | 3,642,940 | 17 | 2.61 | 25 | 21 |
| Texas | 13,724,647 | 1 | 2.865 | 22 | 11.5 |
| Utah | 565,108 | 37 | 1.05 | 38 | 37.5 |
| Vermont | 355,057 | 40 | 2.45 | 26 | 33 |
| Virginia | 4,757,130 | 10 | 2.72 | 24 | 17 |
| Washington | 733,732 | 35 | 0.521 | 44 | 39.5 |
| West Virginia | 3,096,166 | 19 | 8.07 | 6 | 13.5 |
| Wisconsin | 8,814,261 | 2 | 7.07 | 7 | 4.5 |
| Wyoming | 990,703 | 34 | 8.61 | 4 | 19 |

[1] The "Mean Rank" is the average of the number killed rank and the rate killed rank.

Mean deer kill rates vary from 0.049/100 population in California, to a high of 10.7/100 population in Montana. Although the annual deer kill mean in Montana is relatively low (104,845/year, ranked 24th), the state's small human population (~980,000) contributes to the high deer kill rate. In contrast, the low deer kill rate in California is a combination of relatively few deer taken on average (18,087) and a very high mean human population (~37.3 million).

In addition to Montana, the following states have high deer kill rates: Alabama (7.07), Arkansas (6.16), Mississippi (8.98), North Dakota (9.46), South Dakota (8.23), West Virginia (8.07), Wisconsin (7.07), and Wyoming (8.61). Of those states, Alabama (7,338,314), Mississippi (5,583,753), and Wisconsin (8,814,261) were also among the top ten states with the largest *number* of deer killed. Notably, Texas, which had the largest number of deer killed over the study period (N = 13,724,647), ranks 22nd in the deer kill rate once kills are adjusted to reflect human population.

The last column in Table 2 combined the number and rate killed outcomes into a ranked average. Taking this average ranking into account, the top ten states for deer taken were as follows: (1) Wisconsin; (2) Mississippi; (3) Alabama; (4) Michigan; (5) Missouri; (6) Georgia and South Carolina; (7) (skipped); (8) Pennsylvania and Texas; (9) (skipped due to tie for 8th); (10) Arkansas.

*4.3. Results, Deer Kill Hot/Cold Spot Maps*

Figure 1 shows the hot/cold spot map for number of deer killed, while Figure 2 shows hot/cold spot map for the deer kill rate. Figure 3 shows hot/cold spots maps for deer kill rates using deer population estimates as the base for one year, 2017, the only year for which these data were available. These three figures provide a visual representation of different dimensions of deer killing. Each figure presents data based on measures of the standard deviation from the mean. Here, we used the means for the time period for each state. It would be possible to create separate annual maps for all states each year, which would produce 22 additional maps for analysis. Those maps would help provide information

about the potential change in the deer taking distribution over time. That level of detail is beyond the scope of this initial examination of deer hunting distribution in the US, which has not been previously examined. Such analyses may be appropriate for future research and would be relevant if our research question sought to address the nature of changes in deer hunting in each US state.

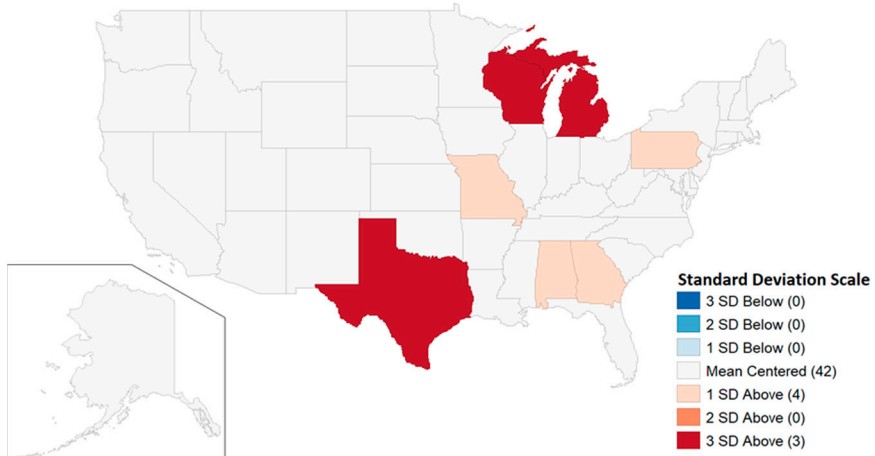

**Figure 1.** Standard deviation (SD)[1] map, total number of deer killed, US, 1999 to 2020. Standard deviations above and below the mean. Number of states in each category in parentheses.

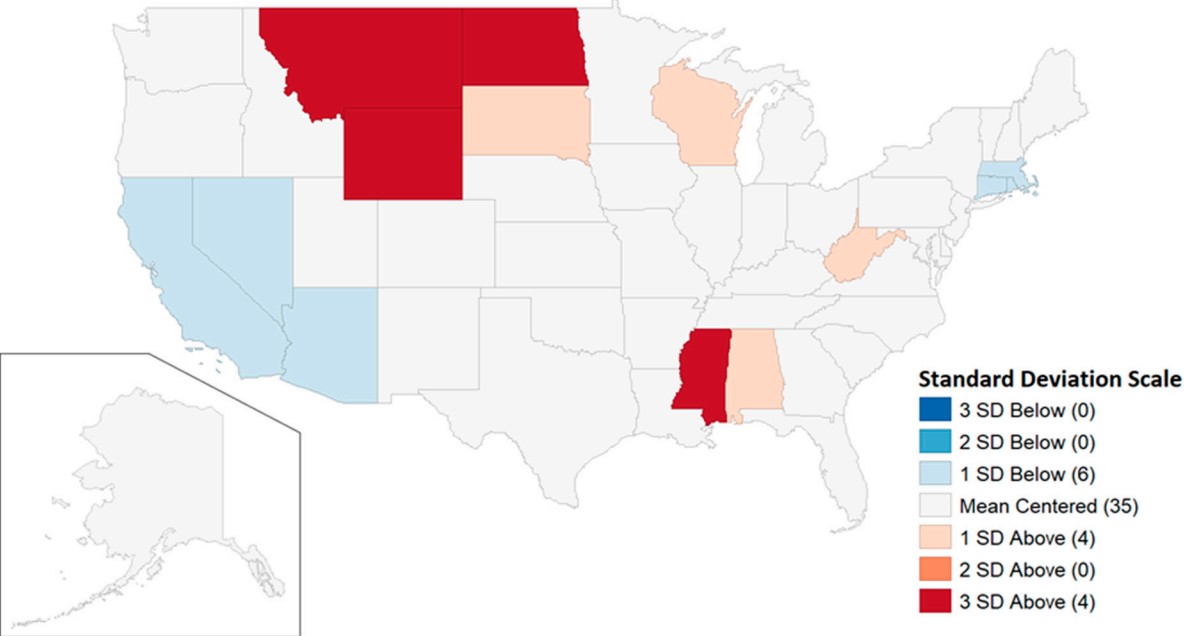

**Figure 2.** Standard Deviation Map, Deer Killed Rate, US, 1999 to 2020. Standard deviations above and below the mean. Number of states in each category in parentheses.

For hot/cold spot maps, we plotted standard deviations from the mean. Statistically, two standard deviations from the mean is considered statistically significant. Figure 1 indicates that Texas, Michigan, and Wisconsin were hot spots for number of deer taken (i.e., more than two standard deviations above the mean). Note that no states were one standard deviation or more below the mean—that is, there were no extreme deer taken cold spots for number of deer harvested.

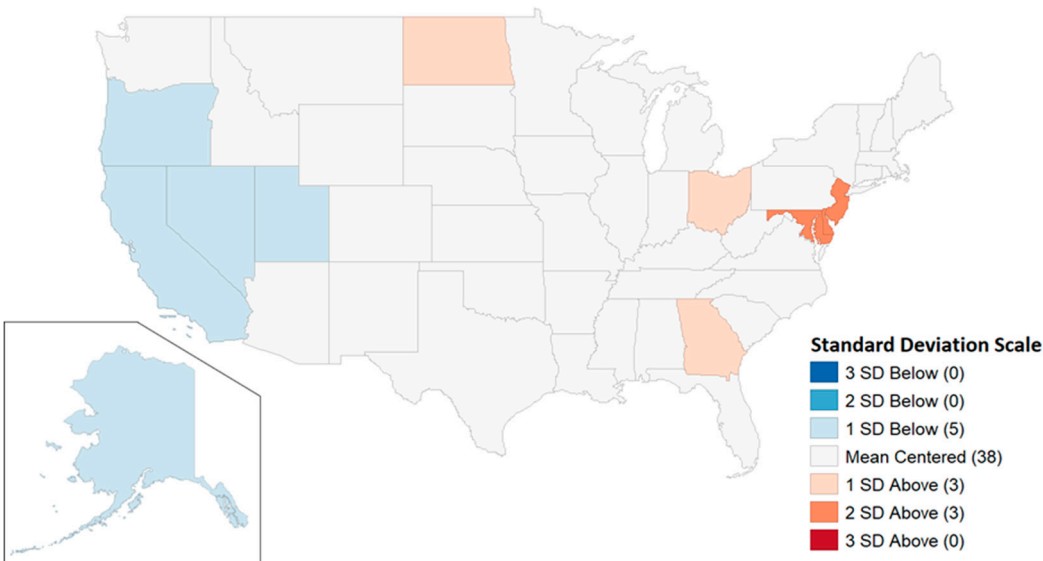

**Figure 3.** Standard deviation map (SD)[1], deer kills to deer population ratio, US, 2017. Standard deviations above and below the mean. Number of states in each category in parentheses.

When analyzing deer kill rates relative to human populations, we observe a different pattern. For kill rate, states well above the mean (three or more standard deviations) include Montana, Wyoming, North Dakota, which form a geographic cluster, and Mississippi. Four states (South Dakota, Wisconsin, West Virginia, and Alabama) are between one and two standard deviations above the mean. We also see two low kill-rate state clusters, one in the western US (California, Nevada, and Arizona) and one in the northeast (Massachusetts, Rhode Island, and Connecticut).

Figure 3 examines kills rates using deer population counts for 2017 as the denominator. This is the only year deer population estimates are available for each state. Figure 3 shows two clusters: a cluster below the mean in the western US (California, Nevada, Oregon, and Utah), and a cluster two standard deviations above the mean in the eastern US (New Jersey, Maryland, and Delaware). Three other states have deer kill rates more than one but less than two standard deviations above the mean: Georgia, Ohio, and North Dakota.

Across all figures, the geographic image of deer killing is unclear. The results depend heavily on the metric used to measure deer kills (i.e., number, rate per human population, rate per deer count). The only consistent result is the appearance of California and Nevada among the low deer kills maps regardless of the measure.

Above, while we analyzed the available state level data, lower-level, disaggregated data (e.g., county level) would be an appropriate level of analysis for examining the impacts of within state deer hunting policies on deer taking and deer populations. It is possible that deer hunting rates not only vary across states, but within states as well, and we will comment on this possibility further in the discussion/conclusion.

### 4.4. Results, Predicting the Distribution of Deer Killing

The correlation between the mean deer kill rate and mean household income across states was negative (r = −0.466; r² = 0.217) and significant ($p < 0.01$). This relationship indicates that across states, deer kills increase with lower household income. This outcome supports the possibility that people in states with lower incomes have a greater proportion of people who hunt for subsistence. This is a potentially important finding that indicates some need to consider subsistence need hunting in the construction of deer hunting policy. The correlation between mean gun ownership and the deer kill rate was positive and significant (r = 0.577; r² = 0.333; $p < 0.01$). This relationship indicates that more deer are killed in states with higher levels of gun ownership. The last correlation between the rate of gun homicides and the deer kill rate across states was in the expected, positive direction,

but was statistically insignificant (r = 0.162; r$^2$ = 0.026), suggesting that gun homicides and deer killing are unrelated, and that deer killing does not appear to be a general expression of what can be called a "gun violence culture."

## 5. Discussion

Green criminology examines the ways humans harm various parts of the ecosystem, commit crimes against nature, and how ecological harms generated by humans can be controlled. One portion of that literature specifically involves theories and studies of harms against nonhuman animals. These studies, following Beirne [11], argue that the killing of animals deserves its own term—theriocide—to illustrate that those killing are important social and criminological concerns.

Although green criminologists have drawn attention to crimes/harms to animals, and to the normalization or legal killing of animals, they have not specifically examined the empirical extent to which those harms occur in society. Criminologists spend considerable time and effort counting crimes that harm people—what are often called street crimes—but not crimes that harm animals. Here, we drew attention to that deficit in the literature by examining the extent of legal deer killing (i.e., theriocide) in the United States over the past two decades. As noted, large numbers of deer were killed (more than 142.3 million) during the study period, and there was considerable variability in deer killings across US states. We showed that deer killings were related to mean household income across states (i.e., states with lower mean household incomes had higher deer killing rates), and gun ownership rates (i.e., deer killing was positively related to gun ownership rates). We also noted that there is a long history of deer hunting in the US, and that the culture of deer hunting may vary across states. That history may affect levels of deer hunting, but empirically, the construct "deer hunting culture" is difficult to measure. Furthermore, observations contained in the extant literature indicate that the taking of deer may also be influenced by other social and political factors such as the number of interest groups involved in policy-making decisions concerning deer hunting and the control of deer populations in different locations. What should be made of these observations and results?

As green criminologists, and drawing on relevant philosophical arguments in that literature, it is tempting to conclude that deer killing is, in and of itself, a harm that ought to be prevented. To be sure, in an ideal world, the preservation of the life of any species ought to be a strong motivating, philosophical rationale for preventing harms. Although that might be ideal, we recognize that other factors may affect how deer hunting is viewed. Importantly, for example, the health of ecosystems and forests in the US today has been adversely affected by human activity over hundreds of years. At this time, then, the current conditions in forest lands in different places may make a ban on deer hunting unfeasible. Accepting this conclusion requires some explanation.

Today, natural ecosystems are smaller than they once were, and they are much more constrained, strained, and impeded in many ways. The effects of humans on ecosystems have changed their ability to function as they once did. With respect to deer populations, for instance, earlier in history, the expansive intact forested land covering the US could support larger deer herds. For example, according to the US Forestry Service, there were about 1,050,000 million acres of forest in the US in 1760 compared to about 745 million today, or more than 305 million fewer forest acres. Estimates of the number of deer that can be supported on different kinds of forested land vary from 1:8 to 1:30 acres. Given these figures, 305 million forest acres could support between 10.2 to 38.1 million deer. Not only has the volume of forest contracted, but it has also been replaced in some locations by monocultured "forest", and segmented and fragmented by deforestation, logging, roadways, and smaller suburban and urban wood-scaped communities [80,81], impeding the ability of US forest ecosystems to maintain wildlife. These historically generated conditions provide less space for deer and increase the likelihood of human–deer conflicts [82], especially those related to more serious outcomes such as deer-related car crashes. For some interest groups, such as farmers, deer presence is considered a nuisance, and a dimension of human–deer

conflict related to agricultural food and cattle resource security [55]. Deer presence can also interact with and adversely affect efforts to maintain rare plant species [58]. In addition, earlier in history, some portion of the deer population was managed by the activities of predatory animals. The presence of large predator animals, however, has diminished or been eliminated in many places in the US, which was fueled by congressional funding of methods to eliminate predator animals beginning in 1915 [83].

These conditions have created circumstances in which deer are considered overabundant in many US forest systems [8], even in suburban and urban areas [28]. Research indicates that deer overabundance in a forest system can have extreme detrimental effects [66,69,84]. These observations suggest that choosing between preserving deer for philosophical/humanitarian reasons and harvesting deer to protect the cohesion/effectiveness of already compromised ecosystems presents a complex issue that is not easily settled. Moreover, there are other matters noted above, such as the tension between the preservation of rare plant species and deer presence that also need to be considered in efforts to preserve/manage deer populations.

To be sure, this conflict between deer presence and ecological system health need not be solved solely by deer hunting/management. It remains possible, for instance, that the return of predators to ecosystems or the use of deer contraceptive measures might work [72]. Nevertheless, even if hunting deer could be eliminated, this strategy overlooks the fact that there may be low-income populations and indigenous peoples who rely on hunting deer for dietary protein, and that some allowances for deer hunting may be necessary for some communities or peoples.

The observations above present counterpoints to the more expected green criminological frame of reference that would support the elimination of deer hunting today. Here, we have not chosen to promote one side of the argument over another. To a large degree, making that choice involves taking a moral or philosophical position about these different viewpoints, and at the same time taking scientific information into account. There is, in our opinion, no single solution to this question at this point in time. This, we would argue, is a consequence of the nature of the issue itself, and how addressing deer hunting and harms to deer and ecosystems involves an intersection of moral/philosophical and scientific perspectives that cannot be easily integrated or rectified, or ordered (e.g., what should count more, moral issues or scientific outcomes?).

One point we wish to clarify, however, is that when making arguments about the scope of animal harms in society, it is important to provide empirical evidence pertaining to that argument. To do so, green criminologists, for instance, must be willing to discover ways to count the number and types of different species harmed, and cannot continue to settle for making the argument that killing animals is *a facie eius* (i.e., on the face of it) harm or a crime. For example, before humans became settled and learned how to plant and raise crops, they relied upon activities that included hunting for survival. One might wish to think that, in today's world, we are no longer under those same circumstances. Yet, at the same time, about 1.9 billion people globally live in extreme poverty (i.e., on less than the US equivalent of USD 1.25/day, the United Nation's standard for extreme poverty). Those people—including many indigenous peoples in lesser developed nations (or, one could say, people living in purposefully underdeveloped nations that benefit the advancement of developed nations)—may have no real alternatives between hunting/killing animals and survival. Recognizing such possibilities makes it even more difficult for green criminologists in Western/developed nations to suggest that the killing of animals is absolutely, morally wrong everywhere or always. This is an area of work that has yet to unfold in the green criminological literature, and we await further discussion of these points by others willing to step into this difficult debate [85].

The aggregate data for the US employed in this paper has several limitations. First, these observations are limited to the US alone and therefore cannot be generalized to other nations. Second, given the level of aggregation, these data cannot be employed at finer levels to address deer hunting policy effects. Several examples of studies employing more

disaggregated data can be found in the literature, which also address the effects of deer on ecosystems outside the US [86–90]. Third, as noted, these data are likely rough estimates of deer harvested via hunting, and it is difficult to suggest that these data can be applied in more precise ways that have predictive validity.

In the above, we limited the scope of our analysis to address what we interpret as a conflict concerning either moral/philosophical, deer management, or scientific/ecological perspectives toward deer hunting in the US. We did not, as a result, endeavor to address several other questions that might also be examined with these data, and the addition of explanatory variables to the data. For example, an unaddressed question, as one reviewer noted, is whether there is a relationship between the number of hunting licenses sold and the number of deer killed. That is an interesting question that was beyond the scope of our analysis. Another question related to whether the impacts of deer hunting can be seen in the ecological measure of forest integrity and health across states and over time. One can think of many other questions a dataset of this nature might address that are beyond the scope of the present work.

**Author Contributions:** Conceptualization, M.J.L.; Methodology, L.J.G.; Formal analysis, L.J.G.; Data curation, M.J.L. and L.J.G.; Writing—original draft, M.J.L. and L.J.G.; Writing—review & editing, M.J.L. All authors have read and agreed to the published version of the manuscript.

**Funding:** This research received no external funding.

**Data Availability Statement:** Data available upon request.

**Conflicts of Interest:** The authors declare no conflict of interest.

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
