# Peer review of "Deer Slayers: Examining the Scope of and Arguments for and against Legal Deer Theriocide in the US"

_sustainability, doi:10.3390/su15075987_

Round 1

Reviewer 1 Report

This review is from an ecologist with no significant research experience in green criminology, sociology or related fields. Your paper is well constructed and written though maybe green criminology needs to  be defined, in brief, reasonably early in your introduction (say paragraph 3, line 34) to give the article an appeal to the wide readership of Sustainability. Also, perhaps include a short paragraph in the introduction on the link between the (overabundance-related) "deer problem" with the historical and continued  (legal or otherwise) loss of apex predations such as wolves in North America, Europe and beyond.

You occasionally lapse in to rather emotive or subjective language which might detract from the neutral stance of a scientific article: e.g. 'good deal of deer killing' [line 42]; 'violent' [line 69].

The word 'volume' is used frequently (lines 62, 82, 88, 120, 205, 287, 327) with reference to deer killings; I would rather see words such as extent, number or quantity used instead.

line 34: Jones 2015 is not listed in the references

line 45: insert 'or harm' after kill

line 60: excluded (add 'd')

line 97: put Sollund references in date order - 2015, 2016, 2020

line 123: list states in year order, starting with Vermont and ending Arkansas

line 136: suggest 'endeavoured'

line 143: 'improved' (deer herds) - maybe explain/define this word, i.e. does it more numerically more or animals in better condition?

line 150: 'manage' - does this mean conserve/protect from hunting?

line 158:  replace 'ad' with and

line 179: 'presences' perhaps replace with abundance or occurrence

line 180: maybe add 'or damage to commercial forest enterprises' after ecosystems

line 194: maybe insert a reference for deer status in Hawaii

line 198: replace 'except' with 'accept' 

lines 206/7: replace 'not likely to be precise' with 'imprecise'

line 208: 'adjust their reports' - maybe discuss how this is done, here or in the discussion

line 225: delete 'additional' - not needed

line 244: 'the only year for which there are complete' and delete 'data' on next line

line 262: add a reference to show where 'GeoDa' can be found

line 266: delete 's'  in standards

line 279: add 'plus Alaska' after Utah

line 321: insert year (of publication) after 'Beirne'

line 328: maybe replace 'tremendous' with 'considerable'

line 337: you declare your viewpoint/background here - maybe you should do this in the introduction when you first mention the field/concept of green criminology?

line 343:  'places' (add 's')

line 373: maybe replace 'the poor' with 'relatively low income households'

line 385/6: extreme poverty - $1.25/day - maybe state where this metric comes from

References: some of the 'H' authors (Hanberry et al and Hoffmann) are out of order and you are giving a lot of work to a copy editor by using first names of authors rather than initials

Author Response

Reviewer 1. We appreciate this reviewer’s comments and suggestion, and have addressed all of them, we believe, in the manuscript. We provide a brief definition of green criminology; the issue of deer over-abundance is discussed in the manuscript, and some historical context is provided, though this material does not appear in its entirety in the introduction. We have attempted to remove subjected language, and in some cases found this difficult given the nature of the different arguments we are presenting, since some of  those arguments are more subjective. We address use of the term ‘volume’ as suggested, and the majority of other useful language change suggestions the reviewer recommended. We would like to thank the reviewer for these comments, which improved our manuscript.

Reviewer 2 Report

The title is very populistic, misleading and it is not supported by your findings in the manuscript, and also by your line 376 “Here, we are not chosen to promote on side of the argument over another”.

The abstract does not concisely present the aim and the main results; the central conclusion is missing.

The introduction is missing the aim of the paper, and does not fit well with the title and the abstract.

The overall methodology needs to be restructured and clearly presented.

Line 169-173 – it is hard to read and keep track of the numbers. The information will be easier to read in a table.

Only general remarks are included, and no particularities of the current studies are provided.

Results

Line 271-274 - The authors compared the number of deer hunted in the states with the human populations, but there is no data regarding the number of hunters at the US/state level, and also how the number of hunters evolved over time. If the data shown in this manuscript has a long period of monitoring, maybe a comparison between the maximum number of deer hunted in the past years is more appropriate to estimate if the population is or is not declining.

Regarding the 3 maps presented in the manuscript, they look more suitable for the materials chapter than for the results. Basically, the maps are used to visualize some data. They can not be taken as a result.  

Predicting the Distribution of Deer Killing

This chapter is more suited for Methodology or Introduction, than for the Results part.

The discussion chapter needs to be restructured by considering the obtained results, and is not aligned with the purpose of the manuscript, title, and the main results.

Overall, the word “killing” is excessively used, and in my opinion is, also as the title, misleading the readers. Please try to use other general used terms as “hunted”, “harvested”, “legally extracted”, “managed” etc.

 Considering all of these, I consider that this manuscript should be rejected, and I suggest the authors restructure the manuscript and submit it to another journal.

Author Response

Reviewer 2.  Reviewer 2 begins by suggesting that our paper is populistic, and misleading. The reviewer does not define what they mean in any specific manner here, so we had to infer the implication of labeling our work ‘populistic.’. The term ‘populistic’ means to take and support a view that is common among the public. We do not believe our manuscript does this; indeed we believe we do the opposite by reviewing arguments in several different academic literatures, as well as materials on deer management and studies of the effects of deer on ecosystems. We do include some populist arguments, if you want to label them as such, related to the history and importance of deer hunting in the US, and how, in that literature, there is also a discussion of the right to hunt. We do not, however, take a position in our manuscript that supports any particular view as being the absolute answer, and, in fact, view our answer as being far removed from a populace argument. For instance, we refer to the green criminological literature on animal harm as hunting, and note that the popular view in that literature is one that opposes deer hunting. We note that we do not object to that view, but also question whether the absolute perspective that approach lays out applies in all circumstances, such as in the US where the expansion of the deer population has been causing a number of concerning ecological consequences.  We have re-written the abstract to follow more closely the content of our manuscript.

We have edited the introduction.

We believe that information in the text the reviewer found difficult to track is already in the tables. The method is described in parts before the sections to which it applies, except for the portion that relates to the deer count data, which appears before any of the description of the data.

The author would have liked to see the hunting rates calculated based on the number of hunters. When we began collecting these data, that information was unavailable for each state and for each year in our data. But, it should be noted that a beta form of that data has more recently been published (October 2022). We can’t be sure the result would change if the rate was calculated for number of hunters. In any event, we discuss this point in the manuscript, and include a link to the newly released data. Generally, it is well know that as a proportion of the population (as we indicate in the manuscript) engaged in deer hunting has declined over the past two decades, which is the focus of our manuscript. Since our work is not about the efficiency of hunters, or the effect of hunters on deer killing/taking, we do not believe that measuring the deer kill rates relative to hunters has any particular importance to our manuscript, since our manuscript is not about the effect of hunters on deer populations, and whether different size deer hunter populations impact deer population size/abundance. That is an interesting subject but was not the subject of our work. A major point of our work was to address a debate about deer hunting in the literature. In doing so we noted that prior studies have failed to address this point adequately, and lack data on deer hunting, which we noted has not previously been the subject of a manuscript on deer hunting in the US. Our is the first paper to assess long term trends in deer hunting across US states. It is difficult to believe that such basic data has not been examined in this literature, and several assumptions made in this literature cannot be adequately addressed without these data. Having collected these data, we agree with the reviewer that such a deer kill/hunter rate would be useful for certain types if analyzes and discussions, and we attempted to collect those data, but they were unavailable. In planning this study and collecting data, those data were unavailable so we had to proceed in a different way. It would, at this point, now be possible to do an analysis of deer kill rates taking the hunter population into account, but two caveats here should be mentions. First, the hunter data is not specific to the number of deer hunting licenses, but is a count of all types of hunting licenses issued in a state. In some states, large portions of the hunting licenses are for non-deer species, so the available data have limitations with respect to devising this measure. Second, the availability of hunter license counts would also have allowed use to construct the study in an entirely different manner, and address a different set of questions than we explored here.

The reviewer makes an assumption about the effect of hunting on deer populations, which we cannot be sure is true since there is no long-term data in the US by state on deer herd size, which also limited the ways in which the issues we examined here could be explored.

Yes, the maps visualize the data, but they present the data as results from applying a statistical method addressing the difference in standard deviation from the mean. 

We have addressed use of the word “killing.” Note, that in doing so, we discussed potential word choices with deer hunters, and while we realize we stand criticized for being populace by the reviewer, deer hunters call it deer killing, and did not have other terms they used to describe what they do, or the successful outcome of their hunting activity. We substituted two terms, which really are not the equivalent, but changed in many places, the term kill/killed/killing to ‘taken’ or ‘harvested.’ Deer hunters do not seem themselves as managing deer, and many deer are hunted but not killed, and research on this point indicates that most deer hunters do not successfully kill a deer during deer hunting season.

Reviewer 3 Report

This paper falls short of the standard for publication in a peer reviewed journal due to the following reasons:

1.    It fails to present a clear case, from the literature, on whether to hunt or not to hunt deer.

2.    Although the authors have access to a huge dataset on deer hunting kills in the USA territories, their analysis seems to fail to tell a concise story. In addition, the statistical analyses are not convincing. For instance, a test of proportions might have helped to tell the story better. Furthermore, the results of the hotspot analyses are also not convincing. It is suggested that they could use a Getis Ord Statistic for better results.

Overall, this manuscript requires further work before it can be considered for publication

Author Response

Reviewer 3.  The reviewer argues we do not present a clear case about whether deer should or should not be hunted, and that we failed to take a side. It was not our point to take a side. We have include some additional discussion of this issue, and note that we do not believe that there is a definitive conclusion that can be reached. One of the problems in this examination is that this issue has been explored from many different perspectives, and each of those perspectives has different aims. Some align their examinations with a preselected aim (e.g., managing deer so there are sufficient quantities to be hunted), so that literature is not useful for making a decision. The other main literatures are written from two opposing views: a moral/philosophical view, and a scientific view. Rectifying these views in itself would be a tremendously difficult task, and require and extensive logical analysis beyond the scope of the present manuscript.

The reviewer would like us to tell a concise story. We think that part of the point here is that the story is not concise, and cannot easily be simplified. We, for example, used the example of the history of deer hunting in Ohio over a 100 year time period to make that point. For instance, we considered making year-by-year plots for the states, and also comparing year-by-year data by state, which became overly complex, and could not be summarized concisely.

The authors suggests using the Getis-Ord (GO) statistic. That statistic is not recommended for state level analysis. Following ArcGis documentation, the GO is only recommended where a unit of analysis has what can be called 8 close neighbors. This is possible at lower levels of aggregation, where neighbor status for unit can be defined by distance (e.g., at the neighborhood, census tract, count levels). This is difficult for states. Of the 50 US states, 36 do not have neighbors on one side or multiple sides because they are bordered by oceans, the Gulf of Mexico, the Great Lakes, Canada or Mexico, and some states have those non-neighbor conditions on multiple borders. This is why we selected to determine whether hot spots existed using SD differences. We expand a bit on this choice now in the manuscript.

Round 2

Reviewer 2 Report

Dear Authors,

Thank you very much for your time and effort.

Author Response

Thank you for the additional opportunity to revise our manuscript, “Deer Slayers: Examining the scope of and arguments for and against legal deer theriocide in the US.”  We apologize for overlooking the request to move material from the results to the ‘Materials and Methods’ section. As social scientists, we often discuss additional analytic methods  added to a project to further the analysis within the results section, and we overlooked the formatting requirements for you journal’s audience.

We have addressed the issues raised in the latest reviews as follows:

  1. Changed the section we labelled as “Data” to “Materials and Methods.” (p.10).
  2. We moved the deer kill hot and cold spot map methods. These are now found as a subsection in the “Materials and Methods” section (p. 12).
  3. In addition, related to # 2, we blended the results from the separate results section on the deer kill maps into the results section as a subsection which is now found on pp. 16-18.
  4. We moved the deer kill distribution prediction methods to the “Materials and Methods” section. These materials can now be found on p. 12-13.
  5. In addition to # 4, we blended the results from the separate section on deer kill prediction into the “results” section. These materials can now be found on page 18.
  6. We corrected the error in the abstract.
  7. In your comments, you noted that footnotes may be added to the figures and tables. We were not sure what materials you may have wanted added to these tables as footnotes.

Reviewer 3 Report

No comments

Author Response

(The authors gave the same response as above.)
